# Overexpression of FRA1 (*FOSL1*) Leads to Global Transcriptional Perturbations, Reduced Cellular Adhesion and Altered Cell Cycle Progression

**DOI:** 10.3390/cells12192344

**Published:** 2023-09-24

**Authors:** Wuroud Al-khayyat, Jake Pirkkanen, Jessica Dougherty, Taylor Laframboise, Noah Dickinson, Neelam Khaper, Simon J. Lees, Marc S. Mendonca, Douglas R. Boreham, Tze Chun Tai, Christopher Thome, Sujeenthar Tharmalingam

**Affiliations:** 1School of Natural Sciences, Laurentian University, Sudbury, ON P3E 2C6, Canada; walkhayyat@laurentian.ca (W.A.-k.); ndickinson@laurentian.ca (N.D.); dboreham@nosm.ca (D.R.B.); tc.tai@nosm.ca (T.C.T.); cthome@nosm.ca (C.T.); 2Medical Sciences Division, NOSM University, 935 Ramsey Lake Rd., Sudbury, ON P3E 2C6, Canada; jake.s.pirkkanen@gmail.com (J.P.); jdougherty@nosm.ca (J.D.); tlaframboise@nosm.ca (T.L.); 3Medical Sciences Division, NOSM University, 955 Oliver Rd., Thunder Bay, ON P7B 5E1, Canada; nkhaper@nosm.ca (N.K.); slees@nosm.ca (S.J.L.); 4Department of Biology, Lakehead University, Thunder Bay, ON P7B 5E1, Canada; 5Department of Radiation Oncology, Radiation and Cancer Biology Laboratories, Indiana University School of Medicine, Indianapolis, IN 46202, USA; mmendonc@iupui.edu; 6Department of Medical & Molecular Genetics, Indiana University School of Medicine, Indianapolis, IN 46202, USA; 7Health Sciences North Research Institute, Sudbury, ON P3E 2H2, Canada

**Keywords:** FRA1, FRA-1, FOSL1, AP-1, transcription factor, CGL1, adhesion, cell cycle, transcriptomics, CRISPR activation

## Abstract

FRA1 (*FOSL1*) is a transcription factor and a member of the *activator protein-1* superfamily. FRA1 is expressed in most tissues at low levels, and its expression is robustly induced in response to extracellular signals, leading to downstream cellular processes. However, abnormal FRA1 overexpression has been reported in various pathological states, including tumor progression and inflammation. To date, the molecular effects of FRA1 overexpression are still not understood. Therefore, the aim of this study was to investigate the transcriptional and functional effects of FRA1 overexpression using the CGL1 human hybrid cell line. FRA1-overexpressing CGL1 cells were generated using stably integrated CRISPR-mediated transcriptional activation, resulting in a 2–3 fold increase in FRA1 mRNA and protein levels. RNA-sequencing identified 298 differentially expressed genes with FRA1 overexpression. Gene ontology analysis showed numerous molecular networks enriched with FRA1 overexpression, including transcription-factor binding, regulation of the extracellular matrix and adhesion, and a variety of signaling processes, including protein kinase activity and chemokine signaling. In addition, cell functional assays demonstrated reduced cell adherence to fibronectin and collagen with FRA1 overexpression and altered cell cycle progression. Taken together, this study unravels the transcriptional response mediated by FRA1 overexpression and establishes the role of FRA1 in adhesion and cell cycle progression.

## 1. Introduction

Fos-related antigen 1 (FRA1) is a transcription factor encoded by the *FOSL1* gene. FRA1 belongs to the *activator protein complex 1* (AP-1) family of *basic leucine zipper domain* (bZIP)-containing transcription factors that regulate gene expression [1]. The AP-1 transcription factors mediate cellular differentiation, proliferation, and apoptosis in response to a variety of extracellular signals, including growth factors, oxidative stress responses, and inflammatory signals [1,2,3,4,5].

The core AP-1 complex consists of hetero- and homodimers of FOS and JUN families [6]. FRA1 is a member of the FOS family of proteins, which also includes c-FOS, FOSB, and FRA2. The JUN family includes c-JUN, JUNB, and JUND. The JUN subtypes form homo- or heterodimers with FOS members, whereas FOS members form heterodimers with JUN subtypes and are unable to homodimerize [1,7]. Other bZIP-containing accessory proteins of the AP-1 complex include members of the ATF, NRF, and MAF families of transcription factors [6].

The AP1 binding sites consist of *TPA-responsive elements* (TRE; 5′-TGAGCTCA-3′) and *cyclic AMP response elements* (CRE; 5′-TGACGTCA-3′) [8,9]. The AP-1 dimer composition plays an important role in determining which AP-1 binding sites are bound and transcriptionally active [10]. In addition, the AP-1 complex can bind degenerate sequences that are similar but not identical to the TRE and CRE binding sites [11]. Therefore, the AP-1 transcriptional response is complex and depends on the balance between the AP-1-binding subtypes, which are, in turn, regulated by multiple signaling pathways [12]. The most studied AP-1 complex consists of c-JUN/c-FOS heterodimers [13,14]. However, the role of FRA1 in the AP-1 transcriptional response is less established [2].

FRA1 is ubiquitously expressed in most tissues, including the liver, breast, lung, brain, heart, and bone [4,15,16,17,18,19]. At the cellular level, FRA1 is expressed in a variety of cell types, including fibroblasts, endothelial cells, epithelial cells, smooth muscle cells, macrophages, and osteoblasts [4,20,21,22,23,24]. Basal FRA1 expression is low in normal tissue states; however, FRA1 expression can be robustly induced in response to extracellular signals via MAPK/ERK signaling cascades [25,26,27]. In addition, the expression pattern of FRA1 can vary depending on the cell type and the stimulus [25,28]. For example, FRA1 expression is induced in epidermal keratinocytes in response to UV irradiation [29]. Similarly, TGF-β activation promotes FRA1 expression in fibroblasts and osteoblasts [30,31]. In contrast, SMAD4 suppresses FRA1 expression in pancreatic cells [32]. In addition to expression, phosphorylation of human FRA1 at Ser252, Ser265, and Thr230 improves FRA1 activity [33]. Here, phosphorylation prevents FRA1 degradation in proteasomes while also increasing its ability to dimerize with JUN members leading to activation of transcriptional responses [34,35]. Therefore, the downstream cellular effects of FRA1 expression and activity vary depending on the context and cell type. In normal tissues, FRA1 expression promotes a variety of cellular functions, including proliferation, differentiation, apoptosis, migration, and tissue development [36,37,38,39,40].

Abnormal FRA1 expression has been reported in various tumors, and its effect on tumor progression varies depending on the cancer type [12,41]. For example, FRA1 is often overexpressed in certain cancers, such as breast cancer, bladder cancer, liver carcinoma, thyroid tumors, gastric cancer, kidney cancer, osteosarcoma, gastrointestinal carcinoma, melanoma, and pancreatic cancer [36,41,42,43,44,45,46,47,48,49]. In these cancer types, FRA1 overexpression is positively correlated with tumor progression, invasion, and angiogenesis [36,41]. In contrast, FRA1 expression is downregulated in other cancer types, including non-small cell lung cancer and cervical cancer [50,51,52]. In these cancer types, FRA1 overexpression induces tumor-suppressive effects. In addition, we have previously shown that the FRA1 gene (*FOSL1*) was deleted or epigenetically silenced in ionizing radiation-induced neoplastic transformants using the CGL1 human hybrid cell line, a precancerous model of cervical cancer [53,54,55]. Here, reinsertion of the FRA1 gene in tumorigenic variants suppressed tumor formation in vivo. In addition to carcinogenesis, abnormal FRA1 expression has been linked to various inflammatory conditions [56,57,58]. Taken together, FRA1 expression can be both beneficial and detrimental depending on the context, and further research is needed to fully understand its role in various physiological and pathological processes. 

The majority of studies implicating FRA1 overexpression with cancer progression utilize correlational approaches [41]. These tumor models contain several gene alterations that contribute to the cancer phenotype. Therefore, it is difficult to discern the molecular effects of FRA1 upregulation in this context. Thus, the primary aim of this study was to investigate the cellular and molecular effects of FRA1 overexpression using the CGL1 nontumorigenic cell model. The CGL1 cell line was created by the hybridization of HeLa, a tumorigenic human cervical cancer cell, and a nontumorigenic human-skin fibroblast [53,59]. The resulting hybrid CGL1 cells share transcriptional and phenotypic profiles with the parental nontumorigenic fibroblast cells [54,55,60]. More importantly, CGL1 cells do not form tumors when implanted in vivo but can be induced to form HeLa-like tumorigenic phenotype using high-dose radiation exposures [54,61,62]. Therefore, CGL1 cells are preneoplastic and nontumorigenic cells and have been extensively used for studying radiation-induced neoplastic transformation [53,54,55,59,63]. 

In this study, FRA1 was overexpressed in CGL1 cells using stably integrated CRISPR-mediated transcriptional activation (CRISPRa). The CRISPRa system consists of dead Cas9 (dCas9), a mutant form of the *Streptococcus pyogenes* Cas9 enzyme devoid of its endonuclease activity and fused to the potent VP64 transcriptional activator, and a guide RNA (gRNA) designed to target dCas9 to the gene of interest [64,65]. This system upregulates the endogenous gene within physiologically relevant levels, a method that has not been previously utilized in FRA1 studies [66]. Using the CRISPRa system, we have generated FRA1 overexpressing CGL1 cells (CGL1^FRA1^) that demonstrated a 2–3 fold increase in FRA1 mRNA and protein levels. The whole transcriptome RNA-sequencing (RNA-seq) analysis revealed that FRA1 overexpression resulted in 298 differentially expressed genes. Here, the expression of the core AP-1 complex members was dysregulated. Gene ontology analysis illustrated that FRA1 overexpression altered various molecular networks, including transcription-factor activity, regulation of extracellular matrix and cellular adhesion, and a variety of cell signaling processes, including protein tyrosine kinase activity, chemokine signaling, and G protein-coupled receptor activity. Moreover, cell adhesion and cell cycle functional assays were performed to corroborate the RNA-seq results. Indeed, FRA1 overexpression reduced cell adhesion and altered the cell cycle progression. Taken together, this study unravels numerous cellular and molecular changes mediated by FRA1 overexpression.

## 2. Material and Methods

### 2.1. Cell Culture

The CGL1 cells were cultured in Minimum Essential Medium (Corning, 10-010CV; Corning, New York, NY, USA), supplemented with 5% calf serum (Sigma-Aldrich, C8056; St. Louis, MO, USA) and 100 U/mL penicillin-streptomycin (Corning, 30001CI). The human embryonic kidney 293-T (HEK293T) cells were purchased from ATCC (CRL-3216). HEK293T cells were cultured in Dulbecco’s Modified Eagle Medium (HyClone, SH3028501; Marlborough, MD, USA), supplemented with 10% fetal bovine serum (HyClone, SH3039603HI) and 100 U/mL of penicillin-streptomycin. Both cell lines were incubated in humidity at 37 °C with 5% CO_2_. 

### 2.2. Design and Cloning of CRISPRa gRNA Sequences into Lentiviral Transfer Plasmid 

The *Sigma-Aldrich Advanced Genomics* bioinformatics tool was utilized to design CRISPRa gRNA target sequences for activation of the FRA1 gene. The CRISPRa system utilized the dCas9 variant fused to the VP64 transcriptional activator, as described previously [64]. The design criteria included the identification of 20 nucleotide guide sequences complementary to the FRA1 gene preceded by a 5′NGG PAM sequence (requirement for *S. pyogenes* Cas enzyme function). The design tool generated gRNA sequences optimized for gene activation with minimal off-target activity, as described previously [65]. The top three FRA1 gRNA target sequences [**#1:** 5′-GTAACTTCCTCGCCGCGCCC-3′ (−237 to −218 from transcription start site); **#2:** 5′-GTATGGGCAGCTACGTCAGG-3′ (−212 to −193 from transcription start site); and **#3:** 5′-TCGGGACCGACGGGCCAAGG-3′ (−104 to −85 from transcription start site)] were selected for cloning into the third generation lentiviral gRNA cloning backbone vector (Addgene, 73795; Watertown, MD, USA). This plasmid contains BsmBI sites for scarless insertion of the gRNA target sequences. Therefore, the BsmBI overhang sequences were appended to the gRNA target sequences and ordered as single-stranded oligonucleotides from IDT. The forward and reverse oligonucleotides were phosphorylated and annealed *in-house* using a 10 μL reaction volume consisting of 10 μM forward and reverse oligonucleotides, 1 mM ATP (NEB), 1X Kinase Reaction Buffer A (NEB), and 5 units of T4 polynucleotide kinase (NEB). The reaction was completed in a thermocycler using the following parameters: (1) 37 °C for 30 min, (2) 95 °C for 5 min, and (3) temperature ramp down to 25 °C at 0.1 °C /s. The samples were diluted by 250 fold by mixing 2 μL of reaction mix with 498 μL molecular grade water (HyClone, SH30538.LS). Single-step digestion and ligation of the vector and annealed oligonucleotides was completed in 20 μL reaction volume consisting of 100 ng lentiviral gRNA transfer plasmid, 2 μL of diluted and annealed oligonucleotides, 1X Tango Buffer (ThermoScientific), 1 mM DTT (NEB), 1 mM ATP (NEB), 10 units BsmBI (ThermoScientific), and 1 μL T4 Quick Ligase (NEB). The reaction was completed in a thermocycler using the following parameters: (1) 37 °C for 5 min, (2) 23 °C for 5 min, and (3) steps 1 and 2 repeated for 6 cycles; 2 μL of the final ligation product was transformed into NEB Stable Competent *E. coli* cells according to the manufacturer’s instructions and selected for ampicillin resistance (100 μg/mL). Select clones were amplified and subjected to plasmid purification using the GeneJet Plasmid Miniprep Kit (ThermoFisher, K0502), according to the manufacturer’s instructions. Successful insertion of the gRNA target sequences into the lentiviral backbone vector was verified by Sanger sequencing performed by *The Centre for Applied Genomics* (TCAG, SickKids, Toronto, ON, Canada). 

### 2.3. Preparation of CRISPRa Lentiviral Particles via Calcium Phosphate Transfection

HEK293T cells were plated into three 10 cm dishes at 1.0 × 10^6^ cells per dish 48 h before transfection. Cells were ~50–60% confluent on the day of transfection. Two hours prior to the transfection, the media was replaced with 9.0 mL/dish of prewarmed complete media without antibiotics. In a 15 mL tube, the following amounts of third generation lentiviral plasmids were mixed to a final volume of 1.35 mL with molecular grade water: 10 μg pMD2.G (Addgene, 12259), 30 μg psPAX2 (Addgene, 12260), and 40 μg transfer plasmid (Addgene, 61425-dCas9 or cloned CRISPRa gRNA plasmids described above); 150 μL of 2.5 M CaCl_2_ were added to the plasmid mixture and mixed by vortex. Next, 1.5 mL of 2× HEPES-buffered saline (0.05 M HEPES, 0.28 M NaCl, 1.5 mM Na_2_HPO_4_, pH 7.0) were added dropwise while the solution was continuously vortexed. After incubating the transfection solution for 15 min at room temperature, 1 mL of solution was added dropwise into each 10 cm plate. After 20 h post-transfection, the media was replaced with 7 mL of complete media per 10 cm dish. The supernatant containing the lentiviral particles was collected 8 h after the media change. The supernatant was centrifuged at 500× *g* for 5 min at 4 °C to remove cellular debris and cleared using a 0.45 μm syringe filter. The viral solution was aliquoted into 1 mL fractions and stored at −80 °C. 

### 2.4. CGL1 Lentiviral Infection for Producing CRISPRa Mediated FRA1 Overexpression 

CGL1 cells were seeded into a six-well plate in complete media at 100,000 cells per well 24 h before the lentiviral infection. Cells were ~40–50% confluent on the day of the infections. Two hours prior to the infection, the media was replaced with 0.9 mL/well of complete media. The infection was initiated via the addition of 1 mL of lentiviral solution and 100 μL of 20X polybrene solution (8 μg/mL final concentration). After 16 h, the virus solution was removed and each well was incubated with 2 mL of complete media. After 48 h, the cells were passaged and transferred to T25 flasks under antibiotic selection (0.8 μg/mL blasticidin for dCas9 selection; 8 μg/mL puromycin for gRNA selection). The antibiotic selection was maintained for four rounds of passage, at which point the cells were maintained without the selection antibiotic. FRA1-overexpressing CGL1 cells were generated via sequential viral infections. First, dCas9-expressing CGL1 cells were generated (CGL1^dCas9^) and verified via immunohistochemical analysis using a Cas9 antibody (ThermoScientific, 10C11-A12). Next, the CGL1^dCas9^ cells were subsequently infected with three unique lentiviral preparations containing varying gRNA sequences designed for FRA1 overexpression (CGL1^FRA1—#1/2/3^). All assays were performed within 4–7 passages from the initial CRISPRa lentiviral infections. 

### 2.5. RNA Extraction 

Total RNA was extracted from CGL1, CGL1^dCas9^, and CGL1^FRA1^ cells using the TRIzol reagent (Invitrogen, Waltham, MA, USA), as described previously [67,68]. Briefly, cells were seeded in six-well plates with complete media and collected at ~70–80% confluency. Cells were washed with phosphate-buffered saline (PBS) and harvested using 500 μL of TRIzol per well. The samples were transferred into a 1.5 mL tube and mixed with 100 μL of chloroform. The mixture was incubated at room temperature for 15 min and centrifuged at 12,000× *g* for 20 min at 4 °C. After centrifugation, the top aqueous layer containing the total RNA fraction was transferred to a new 1.5 mL tube. The RNA was precipitated with 125 μL of isopropanol. The mixture was vortexed for 15 s, incubated for 10 min at room temperature, and centrifuged at 12,000× *g* for 8 min at 4 °C. The supernatant was discarded, and the RNA pellet was washed with 1 mL of 70% ethanol. The solution was centrifuged at 7500× *g* for 5 min. The supernatant was removed, and the RNA pellet was air-dried for 10 min. The RNA pellet was dissolved in 20 μL of molecular grade water in a thermomixer for 10 min at 37 °C at 1000 rpm. The RNA samples were incubated on ice for 20 min, followed by RNA analysis on the NanoDrop spectrophotometer (ThermoScientific, ND-1000). Absorbance ratios of 260/280 nm >1.8 were considered suitable for downstream applications. 

### 2.6. cDNA Synthesis

Complementary DNA (cDNA) synthesis was performed using the *SuperScript IV First-Strand Synthesis* (ThermoScientific, LS18090050) kit according to the manufacturer’s instructions, with minor modifications. Briefly, 1 μg of RNA was subjected to DNase treatment and converted to cDNA using random hexamers and the reverse transcriptase enzyme. The reaction volume was adjusted to 25 μL to obtain a final cDNA concentration of 40 ng/μL and stored at −80 °C. 

### 2.7. RT-qPCR

Forward and reverse primer pairs for SYBR-Green-based RT-qPCR analysis were designed *in-house* and validated under stringent conditions (efficiency between 90–110% and R^2^ > 0.99), as described previously [55,69,70,71]. Table 1 lists the validated RT-qPCR primer sequences utilized in this study. RT-qPCR reactions were performed using the QuantStudio 5 Real-Time PCR instrument (ThermoScientific) in 96-well plates. The RT-qPCR reaction mix was completed in 15 μL reaction volume consisting of 5 ng cDNA, 600 nm forward and reverse primers, and 7.5 μL 2× Luna Universal qPCR Master Mix (NEB, M3003). The RT-qPCR cycling parameters consisted of 95 °C for one min followed by a two-step denaturation and extension cycle of 95 °C for 15 s and 60 °C for 30 s for a total of 40 cycles. Plate reading was performed at the end of the extension phase. A DNA melt curve was performed at the completion of each RT-qPCR experiment to assess the amplification specificity. The RT-qPCR data was analyzed using the 2^−∆∆Ct^ analysis method, as described previously [72,73]. All samples were normalized to two independent housekeeping genes (*RSP18* and *RPL4*). The relative mRNA expression of each gene was reported as the mRNA fold change ± standard error of means (SEM) relative to the CGL1^dCas9^ control cells. 

### 2.8. Protein Extraction and Western Blot Analysis

Cells were harvested in T25 flasks at ~70–80% confluency for protein extraction. Briefly, cells were washed with PBS, trypsinized, neutralized with media, and centrifuged. The cell pellet was washed with ice-cold PBS, followed by resuspension of the pellet with 100 μL of fresh RIPA lysis buffer (150 mM NaCl, 5 mM EDTA, 50 mM Tris, 1% NP-40, 0.5% sodium deoxycholate, and 0.1% SDS; pH 7.5) supplemented with a protease inhibitor mix (ThermoScientific, A32955). The lysis mixture was agitated for 30 min at 4 °C with intermittent vortex, followed by centrifugation at 20,000× *g* for 20 min at 4 °C. The soluble protein fraction was transferred to a 1.5 mL tube and stored at −80 °C. Protein concentration was determined using the Pierce BCA Protein Assay Kit (ThermoScientific, PI23225) according to the manufacturer’s instructions. 

The protein samples were analyzed via gel electrophoresis using the *BOLT Bis-Tris Plus* gel system (ThermoScientific). Briefly, samples were prepared for gel electrophoresis in 50 μL of reaction volumes consisting of 25 μg of RIPA solubilized protein sample, 1X Bolt LDS Sample Buffer (LSB0007), and 1X Bolt Reducing Agent (LSB0004). The samples were sonicated at 10 Hz for 20 s, followed by 70 °C for 10 min. The protein ladder (ThermoScientific, 26619) and samples were loaded onto a *Blot Bis-Tris Plus Mini Gel* (ThermoScientific, NW04120BOX) and electrophoresed for 22 min at a constant 200 V. 

The contents of the gel were transferred to a nitrocellulose membrane (PALL, 66593) using the Mini Bolt Module (B1000) according to the manufacturer’s instructions. The blots were blocked with 5% bovine serum albumin (BSA) in TBS-T buffer (Tris-buffered saline with 0.1% Tween-20) for 60 min at room temperature. The blots were washed three times with TBS-T for 5 min each at room temperature, followed by incubation with the following primary rabbit monoclonal antibodies overnight at 4 °C: FRA1 (1:1000; Cell Signaling #5281T), cFos (1:500; Cell Signaling #2250S), cJun (1:1000; Cell Signaling #9165S), Integrin α4 (1:1000; Cell Signaling #8440T), Integrin α11 (1:500; Cell Signaling #37815S) and Integrin β8 (1:500; Cell Signaling #88300S). The blots were washed four times with TBS-T and incubated with HRP-conjugated goat antirabbit secondary antibody (ThermoScientific, G21234) at 1:10,000 dilution for 1 h at room temperature. After four washes with TBS-T, the blots were incubated with 2mL of ECL mixture (ThermoScientific, 0032109) for 5 min followed by signal imaging using the ChemiDoc Imaging System (Bio-Rad, Hercules, CA, USA). The antibodies were stripped from the blots using a mild stripping buffer (0.2 M glycine, 0.5% SDS, and 1% Tween-20; pH 2.2) for 10 min at room temperature. The blocking, washing, and antibody incubation procedure described above were repeated for the mouse monoclonal Gapdh (1:20,000; ThermoScientific, MA515738) or α-Tubulin (1:1000; Cell Signaling #3873S) housekeeping antibody, followed by the HRP-conjugated goat anti-mouse secondary antibody (1:10,000; ThermoScientific, LSG21234). The band intensities were quantified using the ImageJ analysis software, as described previously [74,75]. 

### 2.9. Cell Growth Assay 

In T25 flasks, 100,000 cells were seeded with complete media. Cells were collected at 24 h intervals for seven consecutive days in duplicates. Media change was performed on day 3 and day 5. Cell collection included a PBS wash followed by the addition of 0.05% trypsin and normalization using complete media. The total cell count per T25 flask was determined using a hemocytometer. The mean cell count for three independent experiments ± SEM is provided for each cell line. 

### 2.10. RNA-seq Whole Transcriptome Analysis

Total RNA was extracted using the TRIzol method and further purified using the NEB Monarch^®^ RNA cleanup kit (NEB, T2040L). Purified RNA (A260/280 ~2.0 ± 0.1) was subjected to RNA-sequencing library preparation using the NEBNext^®^ Ultra™ II Directional RNA Library Kit (NEB, E7765S). This kit purifies poly-A mRNA products and prepares strand-specific directional libraries using the dUTP method for strand-specificity. The library-prepared samples were quantified using the NEBNext^®^ Library Quant Kit (NEB, E7630S). The samples were pooled at equimolar concentration and sequenced at TCAG on a NovaSeq 6000 Illumina platform at approximately 35 million reads per sample (paired-end 100 base-pair reads). The sequencing data was processed *in house* using various bioinformatics toolkits offered by *DRAGEN Inc.* via the *Illumina Sequence Hub* (San Diego, CA, USA) platform. Briefly, *DRAGEN FASTQ* was used for sequencing the data quality check and read trims. *DRAGEN RNA* was used to align reads to the human reference genome (GRCh38.p13) and perform transcript-count analysis. *DRAGEN Differential Expression* was used to complete differential expression analysis based on the DESeq2 platform. Differential gene-selection criteria were determined based on gene-fold change <−1.5 or >1.5, false discovery rate (FDR) corrected *p*-values < 0.05, and minimum average read count of 30 transcripts per million. *iPathwayGuide* (AdvaitaBio; Ann Arbor, MI, USA) was used to perform gene ontology analysis, as shown previously [55,76,77]. Select targets were verified via RT-qPCR.

### 2.11. Cell Adhesion Assay 

Cell adhesion assay was performed, as described previously, with minor modifications [74]. Briefly, 7.5 ng/μL fibronectin (Sigma-Aldrich, F2006), 5 ng/μL collagen (Sigma-Aldrich, C5533), or 100 ng/μL poly-d-lysine positive control (Sigma-Aldrich, P7280) were plated on flat-bottom 96-well plates and incubated at 4 °C overnight. The plates were washed with wash solution (0.2% BSA in PBS) and blocked with 2% BSA. The cells were collected by trypsinization and resuspended in cell recovery media (serum-free media supplemented with 20 mM HEPES and 0.1% bovine serum albumin, pH 7.4). After 30 min of cell recovery at 37 °C, the cells were plated at 25,000 and 50,000 cells/well for the fibronectin and collagen-coated wells, respectively, and both cell counts were also plated on the poly-d-lysine positive control wells. After one hour, the cells plated on fibronectin and collagen were washed three times with a prewarmed wash solution. All wells were then fixed with 4% paraformaldehyde for 10 min at room temperature, followed by 0.1% crystal violet stain for 30 min at room temperature. The wells were washed with water three times and air-dried. The crystal violet contained in the adhered cells was solubilized using 2% SDS (30 min at room temperature). The solubilized crystal violet was quantified by measuring the ultraviolet absorbance at 550 nM using the Cytation 5 BioTek (Agilent, Santa Clara, CA, USA) plate imager. The absorbance readings for cells plated on fibronectin and collagen were normalized using the poly-d-lysine positive control wells. The relative cell adhesion was reported as fold change ± SEM relative to CGL1^dCas9^ controls. 

### 2.12. Cell Cycle Flow-Cytometry Analysis

In T25 flasks in complete media, 100,000 cells were seeded. The media was replaced 48h later with complete media. Cells were collected at the following timepoints post media change: 2, 4, 8, and 24 h. Cell collection involved a PBS wash, trypsinization, neutralization, cell count using a hemocytometer, and transfer of 250,000 cells/flask to assay tubes. The cells were washed with PBS and fixed with 70% ethanol at −20 °C overnight. The cells were washed with ice-cold PBS and centrifuged at 300× *g* for 5 min. The cells were then resuspended with 300 μL of stain solution consisting of 50 μg/mL of propidium iodide (PI) nucleic acid fluorescent stain (ThermoFisher, P1304MP) and 100 μg/mL of RNAseA (NEB, 7013S) prepared in PBS. After 10 min incubation, the samples were analyzed on a flow cytometer (Sony SA3800). *Kaluza* (London, UK) was used to process the flow-cytometer data in order to identify the proportion of cells that are in one of the three interphase stages of the cell cycle based on the PI fluorescent cell distribution and intensity.

### 2.13. Quantification and Statistical Analysis 

Data were generated from a minimum of three independent experiments per assay and presented as mean ± SEM. Excel (*t*-test) or GraphPad PRISM (one-way or two-way ANOVA) was used to determine significant differences between experimental conditions (*p* < 0.05).

## 3. Results

### 3.1. Establishing FRA1-Overexpressing CGL1 Cells

Stable FRA1 overexpressing CGL1 cells were generated using lentiviral-mediated genomic integration of the CRISPRa components (dCas9 and gRNA targeting the FRA1 promoter). This system resulted in the upregulation of the endogenous FRA1 gene. First, dCas9-expressing CGL1 cells were generated (CGL1^dCas9^) via lentiviral mediated delivery of the dCas9 gene. Next, the CGL1^dCas9^ cells were subsequently infected with three lentiviral preparations containing unique gRNA sequences designed for FRA1 overexpression (CGL1^FRA1—#1/2/3^). A Western blot analysis of the FRA1 protein expression in the CGL1 and CGL1^dCas9^ control cell lines demonstrated that dCas9 expression had no effect on basal FRA1 protein expression as expected (Figure 1A). More importantly, CGL1^FRA1—#1^ and CGL1^FRA1—#3^ showed a 3.0- and 3.5-fold upregulation of FRA1 protein expression, respectively, compared to CGL1 cells (*n* = 3, *p* < 0.05), while FRA1 expression in CGL1^FRA1—#2^ was elevated but not statistically significant (Figure 1A). Therefore, CGL1^FRA1—#3^ was chosen for all further downstream experiments, as this CRISPR activation system demonstrated the most robust FRA1 upregulation and will be referred to hereinafter as CGL1^FRA1^. Furthermore, RT-qPCR analysis was performed to confirm FRA1 overexpression (Figure 1B). CGL1^FRA1^ demonstrated a two-fold increase in FRA1 mRNA levels relative to CGL1^dCas9^ (*n* = 3, *p* < 0.05). Taken together, the CRISPRa system designed for FRA1 overexpression was successfully engineered into the CGL1 cells, resulting in the CGL1^FRA1^ cell line, demonstrating stable upregulation of the endogenous FRA1 gene.

In order to assess whether FRA1 overexpression affected the cell growth pattern, a growth-curve analysis was performed using CGL1, CGL1^dCas9^, and CGL1^FRA1^ cell lines (Figure 1C). Cell growth was quantified at 24 h intervals for seven consecutive days. There were no significant differences in the cell growth patterns for all three cell lines (*n* = 3). In addition, phase-contrast imaging of the exponentially growing CGL1, CGL1^dCas9^, and CGL1^FRA1^ cell lines revealed that there were no observable differences in the cell morphologies between the cell lines (Figure 1D). In addition, all cell lines demonstrated fibroblast-like features. Therefore, FRA1 overexpression had no effect on the cellular growth rate and the morphological features of the CGL1 cells. Additionally, CGL1^dCas9^ was chosen as the control cell line for all downstream experiments, as dCas9 expression had no effect on FRA1 expression, the cell growth pattern, or cell morphology compared to CGL1 cells (Figure 1).

### 3.2. Whole Transcriptome RNA-seq Analysis of CGL1^FRA1^ Cells Relative to CGL1^dCas9^ Cells

RNA-seq whole transcriptome analysis was performed to elucidate the global gene-expression changes induced by FRA1 overexpression. RNA sequencing of CGL1^FRA1^ and CGL1^dCas9^ cells at a depth of 35 million reads per sample resulted in the identification of 13,158 unique mRNA transcripts. Of this, 298 significant differentially expressed genes (DEG) were identified in CGL1^FRA1^ cells relative to CGL1^dCas9^ cells (DEG criteria: fold-change <−1.5 and >1.5 and FDR adjusted *p*-value < 0.05). Figure 2A shows a volcano plot illustrating the 124 upregulated DEGs (red circles) and 174 downregulated DEGs (blue circles). In order to confirm the transcriptome dataset, twelve representative genes were randomly selected from the RNA-seq analysis and cross-verified using RT-qPCR analysis. The genes analyzed included six DEGs (*ITGA4*, *FOSL1*, *CD24*, *ITGβ8*, *FOS*, and *FN1*) and six non-DEGs (*CCND1*, *EGFR*, *JUN*, *ITGB1*, *LAMA1*, and *CTNNB1*). An additional FRA1-overexpressing CGL1 line was assessed (CGL1^FRA1—#1^) in order to screen for potential off- target effects of the CRISPRa gRNA. Figure 1B demonstrated that the mRNA fold change between the RNA-seq and RT-qPCR analyses were similar in magnitude and significance across the twelve representative genes and the two different CGL1 FRA1 overexpressing cell lines (*n* = 3; *, *p* < 0.05). These results provide confidence in the RNA-seq dataset and its use in downstream bioinformatics analyses.

The top 15 upregulated and downregulated genes in CGL1^FRA1^ cells are represented in Figure 2C and Figure 2D, respectively. The tables contain the gene description and *p*-value and are ranked based on fold change. The majority of the top 15 DEGs had a fold change between ±4–10. However, the top two upregulated and downregulated genes had extreme differences in fold change: *KLHL15* (2.3 × 10^6^), *AUNIP* (162.2), *UTP14C* (−6.3 × 10^6^), and *PCDHGB5* (−81.8). The full list of 298 DEGs is presented in Appendix A in alphabetical order. The table provides information on the fold change, *p*-value, and the transcript counts (transcript per million). 

### 3.3. Gene Ontology (GO) Enrichment Analysis of DEGs in CGL1^FRA1^ Cells

In order to classify the DEGs induced by FRA1 overexpression into cellular and molecular pathways, GO enrichment analysis was performed. The *iPathwayGuide* (AdvaitaBio) GO analysis package was used to hierarchically rank the co-occurrence of the 298 DEGs within annotated GO units identified by the GO classification system. The GO analysis included FDR correction to obtain GO terms with increased statistical significance. Table 2 summarizes the top enriched GO terms categorized as molecular functions, biological processes, and cellular components, ranked by q-value (FDR-adjusted *p*-value). The table includes the total number of genes annotated within the GO database and the number of DEGs identified within each GO term. 

The GO molecular function analysis identified numerous GO terms aligned with the established role of FRA1 as a transcription factor: *DNA-binding transcription-factor activity* (GO:0003700), *RNA polymerase II specific DNA-binding transcription-factor activity* (GO:0000981), and *RNA polymerase II cis-regulatory sequence DNA binding* (GO:0000978). Interestingly, the GO analysis identified several molecular functions previously less established with FRA1 activity, such as *extracellular matrix profile and cellular adhesion* (GOs: 0005201, 0005178, and 0005518), and cell signaling processes, including *glycosaminoglycan binding* (GO:0005539), *protein tyrosine kinase activity* (GO:0030296), *chemokine activity* (GO:0008009), and *G protein-coupled receptor signaling* (GO:0001664). The GO biological process analysis consisted of GO IDs related to *positive regulation of multicellular organismal processes* (GOs: 0051240, 0051239, 0032501, and 0050789) and *positive regulation of developmental processes* (GOs: 0051094, 0050793, and 0048731). In addition, biological processes, including *response to external stimulus* (GO:0009605) and *regulation of cell population proliferation* (GO:0042127), were also significantly enriched. Overall, the biological processes identified by the GO analysis are in alignment with the established role of FRA1 and the AP-1 complex in regulating cellular differentiation and proliferation. Finally, the cellular component analysis identified GO IDs that act at the level of the cell membrane or extracellularly: *external encapsulating structure* (GO:0030312), *extracellular matrix* (GO:0031012), *collagen-containing extracellular matrix* (GO:0062023), and *cell periphery* (GO:0071944). In summary, the GO profiles suggest that FRA1 overexpression positively regulates multicellular organismal and development processes by acting on molecular pathways involved in transcription-factor activity, regulation of the extracellular matrix profile and cellular adhesion, and a variety of cell signaling processes that initiate with receptor activation at the level of the cell membrane.

### 3.4. FRA1 Upregulation Alters the Expression of Core AP-1 Complex Members 

The GO analysis revealed that FRA1 upregulation induced a wide range of gene-network changes previously established with AP-1 activity. The AP-1 dimer composition plays an important role in determining which AP-1 binding sites are bound and transcriptionally active. Therefore, the mRNA expression of the core AP-1 complex members (Jun and Fos family genes) was determined in order to understand whether FRA1 upregulation shifts the AP-1 dimer composition (Figure 3A). Indeed, FRA1 overexpression significantly downregulated the mRNA levels of *FOS* (expresses cFos protein) and *JUNB* by 1.53 and 1.49 fold, respectively, relative to CGL1^dCas9^ cells, while the expression of *JUN* (expresses cJun protein) was significantly increased by 1.32 fold (*n* = 3, *p* < 0.05). On the contrary, the expressions of *FOSL2*, *FOSB*, and *JUND* were unaffected by FRA1 overexpression. Similarly, the expression of accessory AP-1 binding partners including the gene members of the ATF, NRF, and MAF families were unchanged between CGL1^FRA1^ and CGL1^dCas9^ cells (Appendix A). In order to assess whether the gene-expression changes translated to protein-expression alterations, select AP1 complex members were analyzed via Western blotting (Figure 3B,C). Similar to the gene-expression results, the protein expression of FRA1 and cJUN were significantly upregulated by 3.0 and 1.5, respectively, relative to CGL1^dCas9^ cells, while the expression of cFOS was significantly downregulated by 3.3 fold (*p* < 0.05). In summary, FRA1 overexpression altered the expression of AP-1 complex members. 

### 3.5. FRA1 Upregulation Alters the Adhesion Profile of CGL1 Cells

The GO analysis of CGL1^FRA1^ cells identified multiple networks across all three GO domains related to the extracellular matrix (ECM) profile and integrin binding, with particular emphasis on collagen-containing ECM and collagen binding (Table 2). Therefore, the mRNA expression of select members of the integrin family and ECM network was identified in order to understand the impact of FRA1 upregulation on the expression of these gene networks. The expression of all integrins detected by the RNA-seq analysis (average read counts >30 transcripts per million) in CGL1^FRA1^ and CGL1^dCas9^ cells is provided in Figure 4A. The expression of *ITGA4* (integrin α4) was significantly upregulated by 2.7 fold in CGL1^FRA1^ cells relative to CGL1^dCas9^, while the expression of *ITGA11* (integrin α11), *ITGB5* (integrin β5), and *ITGB8* (integrin β8) were significantly downregulated by 2.1, 1.4, and 1.6 fold, respectively (*n* = 3, *p* < 0.05). In addition, *ITGB4* (integrin β4) was non-significantly downregulated by 2.2 fold (*p* = 0.051). The expression of the other integrins was unaffected by FRA1 overexpression, including *ITGB1* (integrin β1), which forms heterodimers with the majority of alpha integrins (Figure 4A). In order to confirm whether the gene-expression results translated to protein changes, select integrins were analyzed via Western blotting (Figure 4B,C). Similar to the RNA-seq results, the protein expression of integrins α11 and β8 were significantly downregulated by 4.1 and 2.3, respectively, relative to CGL1^dCas9^ cells, while the expression of integrin α4 was non-significantly increased by 1.4 fold (*p* = 0.10). Taken together, the decreased expression of integrin α11 (collagen receptor) and integrin β8 (fibronectin receptors) suggests that FRA1 upregulation impairs the ability of CGL1 cells to bind collagen and fibronectin.

The expression of select ECM-network genes in CGL1^FRA1^ cells relative to CGL1^dCas9^ cells is presented in Figure 4D. In line with the changes to the integrin profile, the expression of *FN1* (fibronectin 1), *COL14A1* (collagen type XIV alpha 1 chain), and *COL16A1* (collagen type XVI alpha 1 chain) were significantly downregulated by 2.0, 2.8, and 1.4 fold, respectively (*n* = 3, *p* < 0.05). In contrast, the expression of *LAMB2* (laminin subunit beta 2), *COL3A1* (collagen type III alpha 1 chain), *COL4A5* (collagen type IV alpha 5 chain), and *COL8A1* (collagen type VIII alpha 1 chain) were modestly increased by 1.2, 1.5, 1.2, and 1.2 fold, respectively (*n* = 3, *p* < 0.05). In addition, analysis of genes encoding ECM degradation proteins, including *MMP2* (matrix metallopeptidase 2) and *MMP3*, were downregulated by 1.6 and 2.5 fold, respectively. Finally, the expression of the integrin activity modulators *TGFB1* (transforming growth-factor beta 1) and *TGFBI* (transforming growth-factor beta induced) was downregulated by 1.2 and 1.7 fold, respectively. Taken together, the gene-expression changes in the ECM-network profile indicate that FRA1 overexpression reduces ECM expression and ECM-mediated signaling.

The integrin expression patterns (Figure 4A–C) suggest that FRA1 overexpression impairs the ability of CGL1 cells to bind collagen and fibronectin. To test this hypothesis, a cell adhesion assay was performed to assess the ability of CGL1^FRA1^ and CGL1^dCas9^ cells to adhere to 7.5 ng/μL fibronectin and 5 ng/μL collagen (Figure 4E). Indeed, CGL1^FRA1^ cells demonstrated 30% and 35% decreased adhesion to fibronectin and collagen relative to CGL1^dCas9^ cells (*n* = 3; *, *p* < 0.05). In summary, FRA1 overexpression reduced the ability of CGL1 cells to bind collagen and fibronectin, likely via inhibiting the expression of integrin α11 and integrins β5/β8, respectively.

### 3.6. Cell Cycle Analysis of CGL1^FRA1^ and CGL1^dCas9^ Cells

RNA-seq analysis of CGL1^FRA1^ cells relative to CGL1^dCas9^ cells identified numerous DEGs involved in the regulation of the cell cycle (Appendix A): *CDK20* (cyclin-dependent kinase 20; −1.6 fold change), *GAS7* (growth arrest specific 7; −7.5 fold change), *C2CD3* (C2 domain containing centriole elongation regulator three; −1.5 fold change); *EFHD1* (EF-hand domain family member D1; −1.9 fold change), *POLA1* (DNA polymerase alpha 1; −2.7 fold change), *SETBD2* (SET domain bifurcated histone lysine methyltransferase 2; 5.5 fold change), *SMC1A* (structural maintenance of chromosomes 1A; 1.58), and *CENPP* (centromere protein P; 1.56 fold change). Therefore, a cell cycle analysis was performed in CGL1^FRA1^ and CGL1^dCas9^ cells in order to investigate the effects of FRA1 overexpression on CGL1 cell cycle progression. Here, cells grown for 48 h were stimulated with fresh complete media and collected at the following timepoints post media change: 2, 4, 8, and 24 h. Cell cycle phases were determined using PI nucleic acid fluorescent stain analyzed using a flow cytometer. The cell cycle distributions of CGL1^FRA1^ and CGL1^dCas9^ are presented as percent G1 (Figure 5A), G2 (Figure 5B), and S (Figure 5C). Appendix A provides representative cell cycle distribution images for CGL1^FRA1^ and CGL1^dCas9^ samples across the various timepoints. Overall, the cell cycle analysis demonstrated a significant decrease in the distribution of CGL1^FRA1^ cells in the G1 phase of the cell cycle relative to CGL1^dCas9^ cells at the 4, 8, and 24 h timepoints (two-way ANOVA, *p* = 0.0083). However, there were no significant differences between the distribution of CGL1^FRA1^ and CGL1^dCas9^ cells at G2 and S phase. Interestingly, the percent G1 cell cycle results match the gene-expression profile of CGL1^FRA1^ cells, whereby reduced expression of cell cycle genes, such as *CDK20* and *GAS7*, have been associated with increased progression of the cell cycle from G1 to S and G2 phases [78,79]. Taken together, FRA1 overexpression promotes cell cycle progression possibly via transcriptional regulation of genes that promote G1 transition.

## 4. Discussion

FRA1 overexpression has been reported in various pathological states, including tumor progression and inflammation [12,56,80]. However, the transcriptional and functional effects of FRA1 overexpression are still not understood. This is the first study to utilize the CRISPRa system to investigate the cellular and molecular effects of stable overexpression of the endogenous FRA1 gene. Using gene-expression analysis and cellular functional assays, we have unraveled numerous cellular and molecular changes mediated by FRA1 upregulation. First, we have identified 298 significant DEGs in FRA1-overexpressing CGL1 cells. Gene ontology analysis illustrated that FRA1 overexpression altered various molecular networks, including transcription-factor activity, regulation of extracellular matrix and cellular adhesion, and a variety of cell signaling processes, including protein tyrosine kinase activity, chemokine signaling, and G protein-coupled receptor activity. Taken together, these results demonstrate that FRA1 upregulation leads to global transcriptional perturbations. Second, the cell adhesion assay corroborated the RNA-seq results and established that FRA1 overexpression impaired CGL1 cell adhesion to collagen and fibronectin. Finally, we revealed that FRA1 overexpression promoted cell cycle progression via the transcriptional regulation of genes that promote G1 transition. Taken together, this study identified the transcriptional responses mediated by FRA1 overexpression and established the role of FRA1 in cellular adhesion and cell cycle progression.

FRA1 and other members of the AP-1 complex are signal transducers that mediate the cellular response of various extracellular signals. The AP-1 dimer composition plays an important role in determining the gene-expression changes mediated by the extracellular signals. Here, FRA1 overexpression decreased *FOS* expression while *JUN* levels were increased. This suggests that with FRA1 upregulation, there is decreased presence of c-Fos/c-Jun heterodimers and increased FRA1/c-Jun heterodimers. We have previously observed similar shifts in AP1 dimer composition using CO-IP and EMSA techniques in ionizing radiation-induced neoplastic CGL1 transformants with FRA1 re-expression. The gamma-irradiated mutant (GIM) and control (CON) CGL1 transformants were isolated post seven Gy of ionizing radiation exposures. The tumorigenic GIM cells contained deleted or epigenetically silenced *FRA1* gene [53,54,55]. In those studies, reinsertion of the FRA1 gene in tumorigenic variants reduced *FOS* expression while *JUN* expression was increased. Taken together with our current results, we further demonstrate that FRA1 overexpression suppresses *FOS* expression. 

Interestingly, although FRA1 and cFos demonstrate homology in their leucine zipper DNA-binding domains, cFos contains additional transactivation domains required for tumorigenesis and cellular transformation, whereas FRA1 lacks these transactivation domains [4,81]. Perhaps cell types that demonstrate FRA1 upregulation and cFos downregulation demonstrate reduced tumorigenesis. Indeed, we have previously observed that FRA1 re-expression in tumorigenic GIM cells shows decreased cFos expression and the absence of in vivo tumor formation. 

The GO enrichment analysis further emphasized the role of FRA1 as a transcription factor, with multiple GO terms highlighting DNA binding and transcription activity (Table 2). These findings are expected given the structure of FRA1, which contains *basic leucine zipper* DNA binding domains, and previous studies implicating FRA1 with the AP1 transcriptional complex [2,82]. The GO enrichment analysis also identified numerous molecular and cellular GO terms less established with FRA1 activity, especially extracellular matrix profile and cellular adhesion. 

The results from Figure 4 demonstrate that FRA1 overexpression has profound effects on cellular adhesion, including the gene expression of extracellular matrix adhesion proteins and integrins (adhesion receptors). Importantly, we demonstrated that FRA1 upregulation decreased the expression of integrin α11 (collagen receptor) and integrins β5 and β8 (fibronectin receptors). Functionally, these gene changes were consistent with the cell adhesion assay results that revealed a decreased ability of CGL1^FRA1^ cells to adhere to collagen and fibronectin. In contrast, integrin α4 expression was elevated with FRA1 upregulation. In comparison to our previous study on tumorigenic GIM cells, the re-expression of FRA1 also increased integrin α4 expression. Interestingly, integrin α4 does not contain a ligand-binding domain and, therefore, does not participate in cellular adhesion [83]. Integrin α4 is involved in lymphocyte homing [84], suggesting the involvement of FRA1 in immune regulation. In contrast to the current study, re-expression of FRA1 in GIM cells increased integrin α3 and α5 expression and no change in the expression of the other integrins. The differing results in CGL1 and GIM cells demonstrate the effects of FRA1 on integrin expression are cell-type dependent. Indeed, we have previously shown that GIM cells are tumorigenic and display epithelial-like morphology, whereas CGL1 cells are fibroblast-like and are nontumorigenic [53,59,63]. Moreover, there are over a thousand genes differentially expressed between GIM and CGL1 cells, demonstrating that these cell types are transcriptionally dissimilar [53,55]. In addition, there are other studies that also report cell-type-dependent regulation of integrin expression via FRA1 activation. For example, FRA1 represses the transcription of αV and β3 integrin genes in endothelial cells [21], while FRA1 activation induces the expression of integrin α6 in canine kidney cells [85]. 

In addition to integrin expression, Figure 4 illustrates that FRA1 overexpression altered the expression of extracellular matrix genes. Here, the gene expression of fibronectin 1 and collagen subtypes 14A1 and 16A1 were significantly downregulated, while there was a modest increase in the gene expression of collagen subtypes 3A1, 4A5, and 8A1, along with decreased expression of laminin subunit β2. We also identified decreased expression of *TGFB1* and its direct downstream target gene (*TGFBI)* [86]. TGFβ1 promotes the expression of various adhesion molecules, including fibronectin and collagen [87]. Therefore, FRA1 overexpression-mediated downregulation of *TGFB1* may explain the reduced expression of matrix proteins. We also showed that downstream target genes of fibronectin, *MMP2*, and *MMP3*, were downregulated (Figure 4D) [88]. MMPs are matrix metalloproteinases and are involved in the breakdown of extracellular matrix proteins [89]. Given the overall reduced expression of matrix proteins with FRA1 overexpression, the decreased expression of MMP genes is congruent with the overall shift in extracellular matrix genes. Finally, TGFβ1 is a positive regulator of the *epithelial-to-mesenchymal transition* (EMT), and fibronectin is a classic mesenchymal marker [90]. The reduced expression of TGFβ1 and fibronectin with FRA1 overexpression suggests that FRA1 prevents EMT in CGL1 cells. Interestingly, this response seems to be cell-type dependent. In mammary epithelial cells, FRA1 expression promotes TGFβ1 and fibronectin expression, leading to increased EMT and tumorigenic profile [22]. 

In addition to the regulation of cell adhesion, FRA1 overexpression promoted cell cycle progression and the transcription of genes that support G1 transition. First, FRA1 overexpression reduced the gene expression of *CDK20*, a marker for the G1 phase of the cell cycle [78]. Second, FRA1 overexpression upregulated the expression of *STXBP4*, a positive regulator of G1/S cell cycle transition [91]. Third, the expression of genes involved in S and G2 cell cycle phases were modestly upregulated, including *URGCP* and *EVI5* [92,93], while the expression of the growth arrest gene *GAS7* was reduced [79]. Finally, the expression of genes related to the preparation for mitosis was upregulated, including *SETDB2*, *L3MBTL1*, *STIL*, *CENPP*, *ESPL1*, and *EVI5* [94]. Taken together, these transcriptional responses suggest that FRA1 overexpression promotes the exit of cells from the G1 cell cycle phase. Indeed, the cell cycle assay demonstrated that CGL1^FRA1^ cells demonstrated a significant decrease in the distribution of cells in the G1 phase of the cell cycle relative to CGL1^dCas9^ cells (Figure 5). Consistent with these results, there was also a non-significant increase in CGL1^FRA1^ cells at the G2 and S phase of the cell cycle. Similarly, Casalino et al. reported the role of FRA1 in cell cycle regulation and identified cyclin A (*CCNA2*) as a novel transcriptional target of FRA1 [95]. In our study, *CCNA2* was also significantly upregulated by 27% with FRA1 overexpression (FDR *p*-value < 0.05); however, *CCNA2* is not present on the DEG list (Appendix A), as our criteria for gene-fold change was set to <−1.5 or >1.5. Collectively, the gene-expression data and cell cycle assay demonstrate that FRA1 overexpression promotes cell cycle progression.

In addition to cell cycle and adhesion, multiple lines of evidence point towards the role of FRA1 overexpression in immune regulation. For example, the GO enrichment analysis identified *chemokine activity* as a top molecular function. Further review of the RNA-seq DEG list revealed 24 genes that are involved in the immune system. The following genes were upregulated with FRA1 overexpression and are involved in cytokine production or downstream cytokine signaling: *IL6ST*, *CSF2RA*, *IFNGR2*, *CXCL12*, *TNFSF13*, *N4BP2L2*, *TMEM106A*, and *CD177* [96,97,98]. In addition, the majority of immune related DEGs with FRA1 overexpression were downregulated and are categorized as follows: (1) proinflammatory cytokine signaling genes (*CCL2*, *CCL5*, *INAVA*, *TRIM13*, *HERC5*, *IFIH1*, *DDX58*, *CEBPD*, and *NFKBIA*), (2) interferon-specific signaling-pathway genes (*IRF7*, *ISG15*, and *IFI44*), or (3) immune-cell activation genes (*KMT2C*, *NFATC2*, *COLEC12*, and *TCF7L1*) [99,100,101]. The overall trend suggests that FRA1 overexpression mediates anti-inflammatory transcriptional changes in CGL1 cells.

In addition, previous studies have illustrated the role of FRA1 in regulating cytokine production. For example, overexpression of FRA1 in RAW264.7 cells decreased LPS-stimulated IL-6 production [57]. Furthermore, in medullary thymic epithelial cells, FRA1 increased the expression of cytokines CCL-5, CCL-19, and CCL-21, while the expression of IL-1β, IL-6, IL-8, and ICAM1 was downregulated [102]. In terms of inflammatory conditions, FRA1 expression has been primarily associated with increased inflammation. For instance, FRA1 activation enhanced macrophage-mediated inflammation [56]. FRA1 expression is also correlated with the severity of inflammatory bowel diseases [103]. Here, FRA1 knockdown ameliorated inflammation and barrier damage in ulcerative colitis [104]. In contrast to these reports, the gene-expression results from the current study suggest that FRA1 overexpression mediates anti-inflammatory transcriptional changes in CGL1 cells. Therefore, the effects of FRA1 on immune regulation are likely cell-type dependent. 

We and others have previously shown that FRA1 serves as a tumor suppressor in cervical cancer models [53,54,55,59,63]. The CGL1 cells used in this study are a precancerous model of cervical cancer [53,54,55]. Although not the focus of this study, the overall results presented in this study point to numerous lines of evidence supporting the role of FRA1 as a tumor suppressor. First, FRA1 overexpression upregulated the expression of genes involved in DNA damage repair: *KLHL15* [105], *AUNIP* [106], *DNAJC14* [107], *DDB2* [108], and *NFRKB* [109]. Interestingly, *KLHL15* and *AUNIP* were the two highest upregulated genes in CGL1^FRA1^ cells in terms of fold change (Figure 2C). KLHL15 is a ubiquitin ligase that promotes homologous recombination (HR) repair over nonhomologous end-joining (NHEJ) [110], and AUNIP binds to damaged DNA and promotes double-strand break repair [106]. DDB2 is involved in nucleotide excision repair (NER), while DNAJC14 and NFRKB augment the core DNA damage repair systems [111]. The increased basal expression of these DNA-repair genes suggests a tumor-suppressive phenotype. Second, the overexpression of FRA1 downregulated the expression of the following genes involved in transformation and tumor promotion: *WNT9A* [112], *AXIN2* [113], and *MYB* [114]. WNT9a promotes osteosarcoma and its expression is controlled by cFos expression [112]. In our study, FRA1 overexpression suppressed cFos, thereby explaining reduced WNT9a expression. AXIN2 is a scaffolding protein of the Wnt signaling pathway and promotes cancer-cell invasion and metastasis in various types of cancers [113,115]. MYB encodes for a transcription factor that participates in hematopoiesis but is also an established proto-oncogene that drives tumorigenesis [116,117]. In addition, FRA1 overexpression upregulated *EIPR1* (also known as *TSSC1*), a tumor-suppressor gene [118]. *EIPR1* is located in the imprinted gene domain of 11p15.5, an important tumor-suppressor gene region [119]. Third, FRA1 overexpression suppressed the genes involved in EMT (*TGFB1*, *FN1*, etc.) [120]. EMT is a hallmark of cancer progression [121]. Finally, inflammation has a role in promoting tumor progression [122]. A review of clinical data and mouse studies demonstrated that tumor-suppressor inactivation enhanced inflammation and promoted tumor progression [80]. In our study, FRA1 overexpression promoted anti-inflammatory transcriptional changes, further emphasizing the impact of FRA1 in tumor suppression. Taken together, there are multiple lines of evidence supporting the role of FRA1 as a tumor suppressor in CGL1 cells. In future studies, we plan on examining whether CGL1^FRA1^ cells demonstrate altered cellular transformation. The CGL1 cells undergo spontaneous transformation or can be induced to form tumorigenic transformants with high doses (7 Gy) of radiation. Based on the results of this paper, we predict that FRA1 overexpression will reduce the frequency of radiation-induced neoplastic transformation in CGL1 cells. 

In summary, we have unraveled the transcriptional and functional effects of FRA1 overexpression. The results from this study demonstrate that FRA1 upregulation leads to global transcriptional perturbations, reduced cellular adhesion, and altered cell cycle progression. This study also sheds light on the potential roles of FRA1 in immune regulation and tumor suppression. In congruence with the literature, our results further emphasize that the cellular and molecular effects of FRA1 are cell-type dependent. Further research is needed to comprehensively understand the role of FRA1 in cell biology.

## Figures and Tables

**Figure 1 cells-12-02344-f001:**
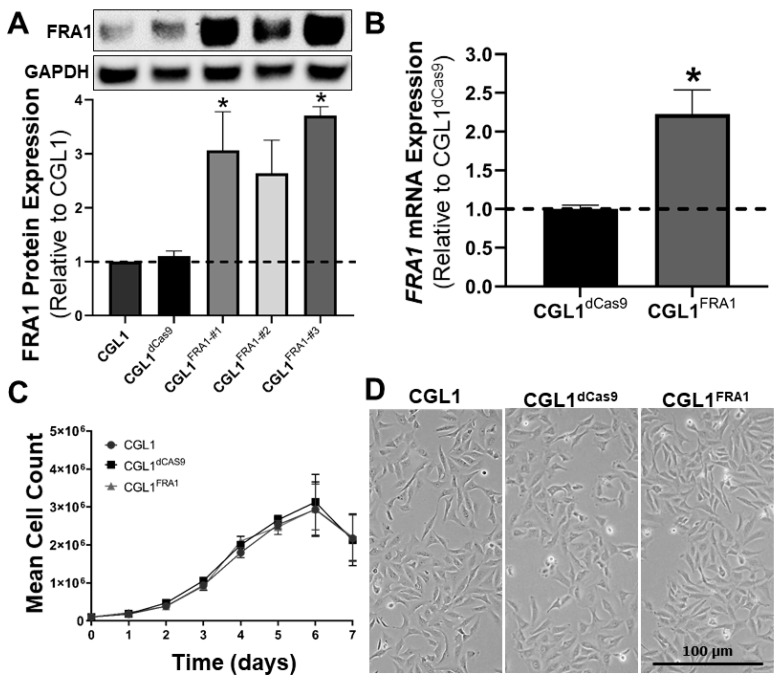
Establishing FRA1 overexpressing CGL1 cells using stably integrated CRISPR transcriptional activation. (**A**) Western blot analysis of FRA1 protein expression in CGL1 and CGL1^dCas9^ control cell lines and three CGL1^dCas9^ variants expressing unique gRNA sequences designed for FRA1 overexpression (CGL1^FRA1—#1/2/3^). Gapdh (glyceraldehyde 3-phosphate dehydrogenase) was used as an internal loading control. Bar graphs illustrate FRA1 protein expression relative to wild-type CGL1 cells (*n* = 3; *, *p* < 0.05). CGL1^FRA1—#3^ was chosen for all further downstream experiments as this CRISPR activation system demonstrated the most robust FRA1 overexpression compared to the other variants. (**B**) RT-qPCR analysis of *FRA1* mRNA expression relative to CGL1^dCas9^. CGL1^FRA1^ demonstrated two-fold increase in *FRA1* mRNA levels relative to CGL1^dCas9^ (*n* = 3; *, *p* < 0.05). (**C**) Cell growth analysis of CGL1, CGL1^dCas9^, and CGL1^FRA1^ cell lines; 100,000 cells were seeded in T25 flasks and the cell growth was quantified at 24 h intervals for seven consecutive days. There were no significant differences in the cell growth patterns for all three cell lines (*n* = 3). (**D**) Representative phase-contrast images of CGL1, CGL1^dCas9^, and CGL1^FRA1^. There were no observable differences in the cell morphology between the cell lines.

**Figure 2 cells-12-02344-f002:**
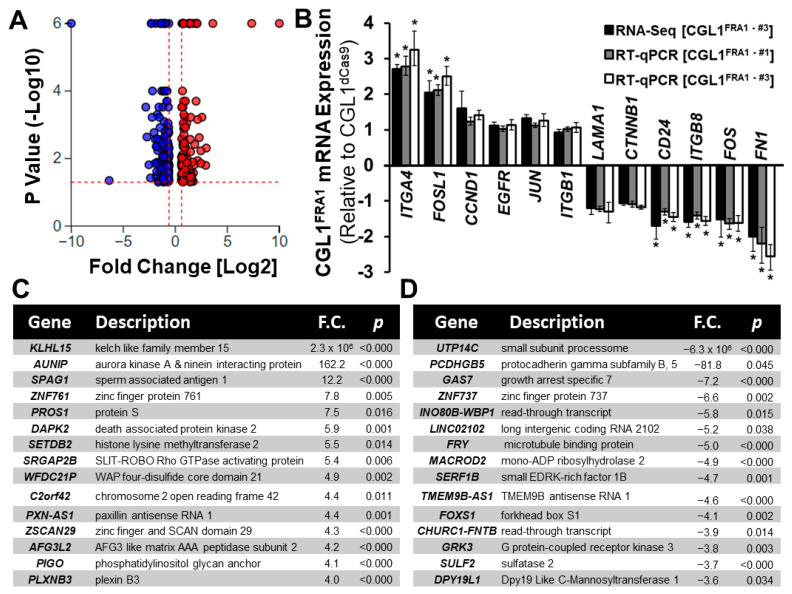
RNA-seq transcriptome analysis of CGL1^FRA1^ cells relative to CGL1^dCas9^ cells. (**A**) A volcano plot illustrating the 298 differentially expressed genes (DEG) between CGL1^FRA1^ and CGL1^dCas9^ cells (DEG criteria: fold change <−1.5 and >1.5 and FDR adjusted *p*-value < 0.05). The red and blue dots represent 124 upregulated genes and 174 downregulated genes, respectively, in CGL1^FRA1^ cells relative to CGL1^dCas9^ cells. The left and right sides of the plot depict highly dysregulated genes, while genes higher on the graph indicate increased statistical significance. (**B**) Validation of RNA-seq dataset using RT-qPCR analysis. Twelve representative genes were randomly selected from the RNA-seq analysis. This list included six DEGs (*ITGA4*, *FOSL1*, *CD24*, *ITGβ8*, *FOS*, and *FN1*) and six non-DEGs (*CCND1*, *EGFR*, *JUN*, *ITGB1*, *LAMA1*, and *CTNNB1*). The graph represents CGL1^FRA1^ gene expression relative to CGL1^dCas9^ (CGL1^FRA1—#1^ and CGL1^FRA1—#3^ represent unique gRNA variants). The mRNA fold change between RNA-seq and RT-qPCR analyses were similar in magnitude and significance across the twelve representative genes (*n* = 3; *, *p* < 0.05). (**C**) Top 15 upregulated genes and (**D**) top 15 downregulated genes identified from the RNA-seq analysis of CGL1^FRA1^ cells relative to CGL1^dCas9^ cells, presented with gene description, *p*-value (*p*), and ranked based on fold change (F.C.). Abbreviations: *CCND1* (cyclin D1), *CTNNB1* (catenin beta 1), *EGFR* (epidermal growth-factor receptor), *FN1* (fibronectin 1), *FOS* (Fos proto-oncogene), *FOSL1* (FOS like 1), *ITGA4* (integrin subunit alpha 4), *ITGB1* (integrin subunit beta 1), *ITGB8* (integrin subunit beta 8), *JUN* (Jun proto-oncogene), and *LAMA1* (laminin subunit alpha 1).

**Figure 3 cells-12-02344-f003:**
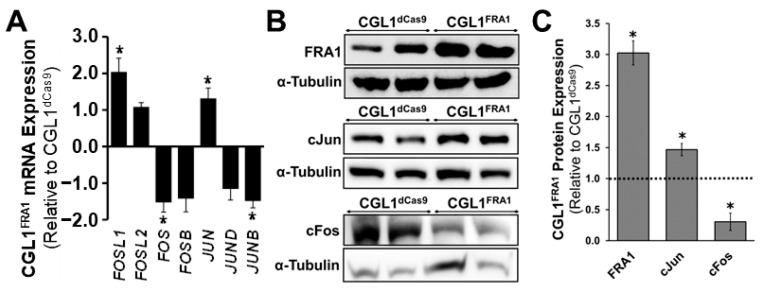
FRA1 upregulation alters the expression of core AP-1 complex members. (**A**) RNA-seq mRNA expression analysis of AP-1 complex gene members. CGL1^FRA1^ gene expression is represented as fold change ± SEM relative to CGL1^dCas9^ cells (*n* = 3; *, *p* < 0.05). (**B**) Representative Western blot images of FRA1, cJun, and cFos protein expression in CGL1^dCas9^ and CGL1^FRA1^ cells. α-Tubulin was used as an internal loading control. (**C**) Bar graphs illustrate CGL1^FRA1^ protein expression relative to CGL1^dCas9^ cells (*, *p* < 0.05). The dashed line indicates CGL1^dCas9^ protein expression. Abbreviations: *FOS* (Fos proto-oncogene), *FOSB* (FosB proto-oncogene), *FOSL1* (FOS like 1), *FOSL2* (FOS like 2), *JUN* (Jun proto-oncogene), *JUNB* (JunB proto-oncogene), and *JUND* (JunD proto-oncogene).

**Figure 4 cells-12-02344-f004:**
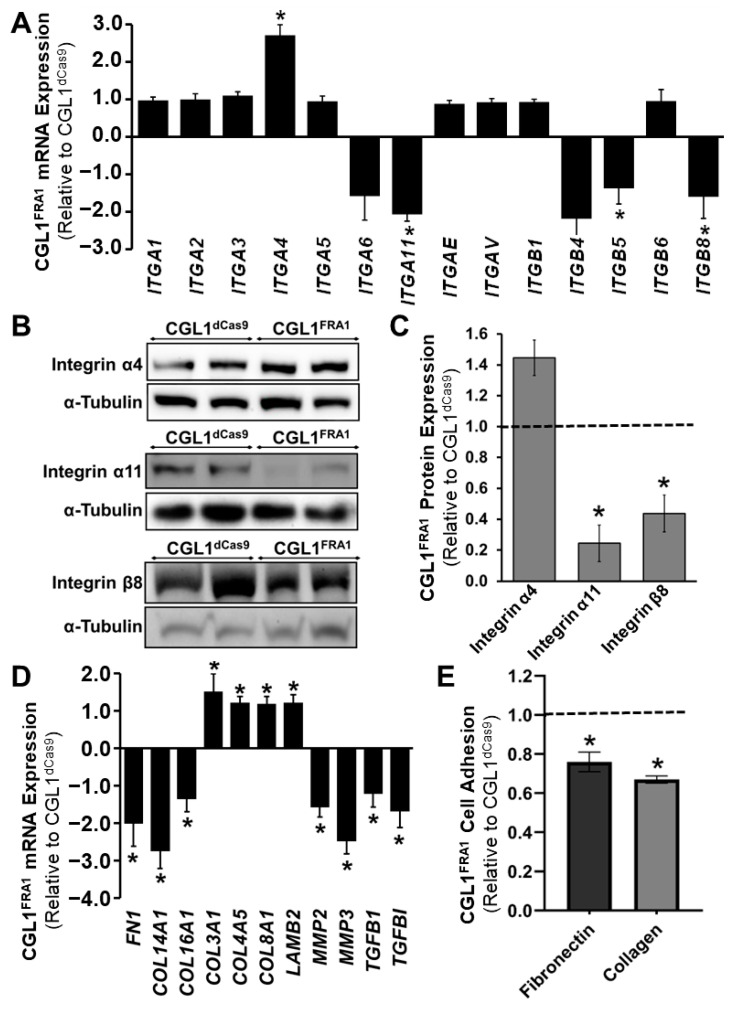
FRA1 upregulation alters the adhesion profile of CGL1 cells. (**A**) RNA-seq mRNA expression analysis of integrin family gene members. CGL1^FRA1^ gene expression is represented as fold change ± SEM relative to CGL1^dCas9^ cells (*n* = 3; *, *p* < 0.05). FRA1 upregulation alters the expression of the ECM profile in CGL1 cells. (**B**) Representative Western blot images of integrin α4, integrin α11, and integrin β11 protein expression in CGL1^dCas9^ and CGL1^FRA1^ cells. α-Tubulin was used in internal loading control. (**C**) Bar graphs illustrate CGL1^FRA1^ protein expression relative to CGL1^dCas9^ cells (*, *p* < 0.05). (**D**) RNA-seq mRNA expression analysis of select extracellular matrix (ECM) network genes. CGL1^FRA1^ gene expression is represented as fold change ± SEM relative to CGL1^dCas9^ cells (*n* = 3; *, *p* < 0.05). (**E**) CGL1^FRA1^ and CGL1^dCas9^ cells were assessed for adhesion to assay plates coated with 7.5 ng/μL fibronectin or 5 ng/μL collagen. The dashed line indicates the adhesion of control CGL1^dCas9^ cells to fibronectin and collagen. CGL1^FRA1^ cell adhesion is represented as fold change ± SEM relative to CGL1^dCas9^ cells. FRA1 overexpression decreased CGL1 cell adhesion to fibronectin and collagen (*n* = 3; *, *p* < 0.05). Abbreviations: *COL14A1* (collagen type XIV alpha 1 chain), *COL16A1* (collagen type XVI alpha 1 chain), *COL3A1* (collagen type III alpha 1 chain), *COL4A5* (collagen type IV alpha 5 chain), *COL8A1* (collagen type VIII alpha 1 chain), *FN1* (fibronectin 1), *ITGA1* (integrin subunit alpha 1), *ITGA11* (integrin subunit alpha 11), *ITGA2* (integrin subunit alpha 2), *ITGA3* (integrin subunit alpha 3), *ITGA4* (integrin subunit alpha 4), *ITGA5* (integrin subunit alpha 5), *ITGA6* (integrin subunit alpha 6), *ITGAE* (integrin subunit alpha E), *ITGB1* (integrin subunit beta 1), *ITGB4* (integrin subunit beta 4), *ITGB5* (integrin subunit beta 5), *ITGB6* (integrin subunit beta 6), *ITGB8* (integrin subunit beta 8), *LAMB2* (laminin subunit beta 2), *MMP2* (matrix metallopeptidase 2), *MMP3* (matrix metallopeptidase 3), *TGFB1* (transforming growth-factor beta 1), and *TGFBI* (transforming growth-factor beta induced).

**Figure 5 cells-12-02344-f005:**
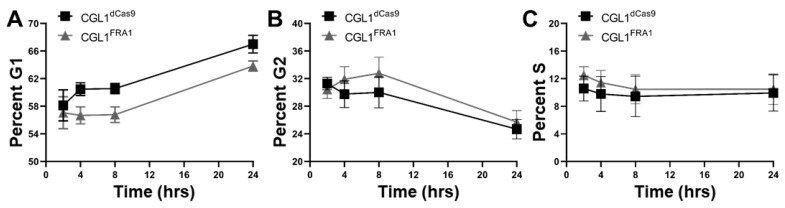
Cell cycle analysis of CGL1^FRA1^ and CGL1^dCas9^ cells. Cells were grown for 48 h followed by a complete media change. Samples were collected at the following timepoints post media change: 2, 4, 8, and 24 h. Cell cycle phases were determined using propidium iodide (PI) nucleic acid fluorescent stain and analyzed using a flow cytometer. The cell cycle distributions of CGL1^FRA1^ and CGL1^dCas9^ cells are presented as percent G1 (**A**), G2 (**B**), and S (**C**). Data points represent mean ± SEM (*n* = 3). CGL1^FRA1^ cells demonstrated a significant decrease in the distribution of cells in the G1 phase of the cell cycle relative to CGL1^dCas9^ cells (two-way ANOVA, *p* = 0.0083). There was no significant difference between the distribution of CGL1^FRA1^ and CGL1^dCas9^ cells at G2 and S phase. Appendix A provides representative cell cycle distribution images.

**Table 1 cells-12-02344-t001:** **RT-qPCR primer sequences.** Forward and reverse primer pairs for SYBR-Green based RT-qPCR analysis designed using Primer-BLAST and validated under stringent conditions (efficiency between 90–110% and R^2^ > 0.99). The optimal annealing temperature for all primer sets is 60 °C.

Gene Name	Gene ID	Primer Seqeunce (5′ → 3′)	
*CCND1*	NM_053056.3	Forward:	ATCAAGTGTGACCCGGACTG
		Reverse:	CTTGGGGTCCATGTTCTGCT
*CD24*	NM_013230.3	Forward:	GCTCCTACCCACGCAGATTT
		Reverse:	GCCTTGGTGGTGGCATTAGT
*CTNNB1*	NM_001098209.2	Forward:	AATCAGCTGGCCTGGTTTGA
		Reverse:	GCTTGGTTAGTGTGTCAGGC
*EGFR*	NM_005228.4	Forward:	GAGCTCTTCGGGGAGCAG
		Reverse:	TCGTGCCTTGGCAAACTTTC
*FN1*	NM_212482.2	Forward:	AACAAACACTAATGTTAATTGCCCA
		Reverse:	TCTTGGCAGAGAGACATGCTT
*FOS*	NM_005252.4	Forward:	GGGGCAAGGTGGAACAGTTA
		Reverse:	AGTTGGTCTGTCTCCGCTTG
*FOSL1*	NM_005438.5	Forward:	GCCTTGTGAACAGATCAGCC
		Reverse:	AGTTTGTCAGTCTCCGCCTG
*ITGA4*	NM_000885.5	Forward:	GCTGTGCCTGGGGGTC
		Reverse:	CACTAGGAGCCATCGGTTCG
*ITGB1*	NM_002211.4	Forward:	GCCGCGCGGAAAAGATGAAT
		Reverse:	CACAATTTGGCCCTGCTTGTA
*ITGB8*	NM_002214.2	Forward:	GGCAGCTGTCTGTGAAAGTC
		Reverse:	CCGTCATTGGGCACCACTAT
*JUN*	NM_002228.4	Forward:	CTTTTCAAAGCCGGGTAGCG
		Reverse:	TTTCTCTAAGAGCGCACGCA
*LAMA1*	NM_000546.5	Forward:	CACTGTTCTGGAAAAGCCCG
		Reverse:	TCAACAAGATGTTTTGCCAACTG
*RPL4*	NM_000968.4	Forward:	CACGCAAGAAGATCCATCGC
		Reverse:	CCGGAGCTTGTGATTCCTGG
*RPS18*	NM_022551.2	Forward:	ATTAAGGGTGTGGGCCGAAG
		Reverse:	GGTGATCACACGTTCCACCT

**Table 2 cells-12-02344-t002:** Summary of gene ontology (GO) enrichment analysis in CGL1^FRA1^ cells relative to CGL1^dCas9^ cells. Top enriched GO terms categorized as molecular functions, biological processes, and cellular components are presented ranked by q-value (FDR-adjusted *p*-value). The table includes the total number of genes annotated within the GO database (Total Genes), and the number of DEGs identified in each GO term (DEGs). GO analysis was performed by *iPathwayGuide* (AdvaitaBio) using the DEGs identified from the RNA-seq analysis of CGL1^FRA1^ relative to CGL1^dCas9^ cells. GO profiles suggest that FRA1 overexpression causes shifts in the extracellular matrix profile, cellular adhesion characteristics, transcription-factor binding, chemokine activity, and regulation of developmental processes.

**GO ID**	**Molecular Functions**	**DEGs**	**Total Genes**	**q-Value**
GO:0005201	extracellular matrix structural constituent	11	89	0.00570
GO:0005539	glycosaminoglycan binding	11	101	0.00975
GO:0000981	DNA-binding transcription-factor activity (RNA polymerase II)	39	863	0.01125
GO:0005178	integrin binding	10	95	0.01125
GO:0030296	protein tyrosine kinase activator activity	4	11	0.01125
GO:0005518	collagen binding	7	46	0.01125
GO:0008009	chemokine activity	3	5	0.01125
GO:0003700	DNA-binding transcription-factor activity	39	894	0.01125
GO:0000978	RNA polymerase II cis-regulatory DNA binding	35	770	0.01125
GO:0001664	G protein-coupled receptor binding	11	122	0.01125
**GO ID**	**Biological Processes**	**DEGs**	**Total Genes**	**q-value**
GO:0051240	positive regulation of multicellular organismal process	45	789	0.00012
GO:0051239	regulation of multicellular organismal process	68	1481	0.00012
GO:0051094	positive regulation of developmental process	43	760	0.00016
GO:0050793	regulation of developmental process	65	1466	0.00049
GO:2000026	regulation of multicellular organismal development	42	783	0.00060
GO:0048731	system development	103	2818	0.00069
GO:0009605	response to external stimulus	62	1406	0.00069
GO:0032501	multicellular organismal process	130	3920	0.00174
GO:0050789	regulation of biological process	199	6873	0.00174
GO:0042127	regulation of cell population proliferation	46	971	0.00257
**GO ID**	**Cellular Components**	**DEGs**	**Total Genes**	**q-value**
GO:0030312	external encapsulating structure	26	267	<0.00001
GO:0031012	extracellular matrix	26	267	<0.00001
GO:0062023	collagen-containing extracellular matrix	19	210	0.00016
GO:0071944	cell periphery	102	2819	0.00072

## Data Availability

The data that support the findings of this study are available from the corresponding author upon a reasonable request.

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
