# Peer review of "Overexpression of FRA1 (FOSL1) Leads to Global Transcriptional Perturbations, Reduced Cellular Adhesion and Altered Cell Cycle Progression"

_cells, 2023, doi:10.3390/cells12192344_

Round 1
Reviewer 1 Report (Previous Reviewer 1)
accept
Author Response
Reviewer #1 had no further specific comments with the updated manuscript and we thank their support for endorsing the manuscript for publication with Cells.
Reviewer 2 Report (Previous Reviewer 3)
The authors improved the manuscript by addressing the reviewer's concerns.
Author Response
Reviewer #2 had no further specific comments with the updated manuscript and we thank their support for endorsing the manuscript for publication with Cells.
Reviewer 3 Report (Previous Reviewer 2)
The authors reported the transcriptome changes in non-cancer CGL1 cells with Fra1 overexpression by CRISPRa. Fra1 has been extensively studied in several cancer/immune cell lines. These studies reported similar results that overexpression of Fra1 regulated cell adhesion, proliferation and immune responses. The unique point of this study is that the authors used non-cancer cell line CGL1 as the cell model. Yet, the study seems preliminary and lacks a solid conclusion. The revision did not improve much of the manuscript. Here are my major concerns:
- Authors emphasized the change of AP-1 composition upon Fra-1 overexpression only based on the down-regulation of FOS protein. How many gene expression alterations is controlled by AP-1 composition shift? Will FOS OE rescue the transcriptome in Fra1 OE cells? Does Fra-1 AP-1 complex bind to different genome locations? Authors need to provide additional analysis.
- For the upstream regulator predictor analysis, the predicted upstream master regulators only regulate a maximum of 13 genes in the prediction table, while RNA-seq indicated 124 up and 174 downregulated genes. More importantly, among these “master regulators”, only FN1 was differentially expressed in Fra1-OE cells. Is there any evidence suggesting the “master regulators” were post-translationally regulated? Given that Fra-1 functions as a transcription factor.
- For the cell cycle analysis, authors reported that Fra1 OE cells showed a decreased proportion of the G1 phase, while the S/G2 phase was unchanged. Overall, no difference was observed regarding cell proliferation. It is uncertain the biological meaning of this analysis.
Author Response
Please refer to attached PDF report.

Round 2
Reviewer 3 Report (Previous Reviewer 2)
I don't have additional comments
This manuscript is a resubmission of an earlier submission. The following is a list of the peer review reports and author responses from that submission.
Round 1
Reviewer 1 Report
In this manuscript, the authors explore the transcriptional and functional effects of FRA1 overexpression via CRISPR-mediated transcriptional activation on the CGL1 cell line. The authors firstly confirm the overexpression level of FRA1 in CGL1 cell line. Then, the authors identified 298 differentially expressed genes with FRA1 overexpression via RNA-sequencing. The authors further discovered that FRA1 overexpression leads to reduced cellular adhesion and altered cell cycle progression via cell functional assays. The authors present lots of bioinformatic analysis and functional assays, however, I think some concerns should be addressed prior to publication in Cells.
Specific comments:
1. In figure 1D, no scale in figure.
2. In figure 3 and figure 4, the author just presents the mRNA expression of these genes, and the mRNA expression difference of most genes is less than 2-fold. So, I suggest the authors also test the protein expression of these genes.
3. In figure 5, lacking the flow image of cell cycle analysis.
Reviewer 2 Report
Al-khayyat and others reported the global transcriptional perturbation induced by endogenous activation of FRA1 in the CGL1 cell line. Authors utilized the cutting-edge CRISPRa technique to overexpress FRA1 in CGL1 specifically, and performed RNA-seq to analyze the downstream effects. Interestingly, authors reported FRA1 overexpression-induced alterations in several biological functions, including cell adhesion, cell cycle progression, immune regulation and etc. Mechanistically, the authors identified the change of AP1 complex composition and performed in silico prediction of upstream regulators. However, the whole study seems preliminary and lacks key experiments to support its conclusion. Here are my major comments:
1. In Figure 3, the authors reported the transcriptional changes of AP-1 complex members caused by FRA1 overexpression. However, the biochemistry evidence supporting this hypothesis is not provided. CO-IP experiments targeting AP-1 complex and changes in protein-protein interaction should be reported. Meanwhile, it remains unclear how the change of AP-1 complex composition alters downstream gene expression. More detailed mechanistic studies should be included, such as motif analysis, downstream gene expression, and binding pattern of AP-1 complex by ChIP.
2. In Figure 4, authors revealed gene expression changes of several ECM networks and integrins in CGL1FRA1. Functionally, CGL1FRA1 demonstrated a reduced cell adhesion property. Besides controlling cell adhesion, integrins and the ECM network also control cellular signaling functions. Does DEG match with the loss of integrin/ECM signaling?
3. In Figure 5, CGL1FRA1 showed an increased proportion of the G1 phase, yet the S and G2 phases were not altered. The authors concluded that FRA1 overexpression enhanced cell progression. However, in Figure 1, the authors reported no difference in cell proliferation and morphology. Authors need to clarify and perform more cell proliferation assays. Also, please provide the original flow cytometry gating and dot plots in Figure 5.
4. Figure 6 is interesting by suggesting some upstream regulators. However, the authors need to provide more evidence to support their conclusion. For example, does gene overexpression/knockout of upstream regulators change DEG? Since only FN1 was found to be regulated in CGL1FRA1, authors conclude other upstream regulators were regulated by post-translational modification, which definitely needs evidence to support.
5. In general, authors concluded that FRA1 overexpression controls several key biological functions, such as immune regulation and tumor progression, based on RNA-seq. Yet, functional experiments need to be provided. Meanwhile, all RNA-seq was performed under a steady state. Does FRA1-overexpression control any of the functions in response to external stimuli? Such as cytokine production and tumor progression.
Reviewer 3 Report
This research article by Al-Khayyat et al. shows the increased expression of FRA-1 in CGL1 cells leads to change in transcriptional profile and various data analysis shows the effect of FRA-1 on cell adhesions and cell cycle progression. In this study Insilco analysis of Bulk RNA-seq predicts important role of FRA-1 in CGL1 cells, however authors did not validated most of these changes and there is lack of functional data.
Authors must show what its the functional out come of impaired cell adhesions and perturbed cell cycle progression in context to cellular transformation and migration.
Authors should also validate more DEG in each category of their analysis to reach that conclusions.
Round 2
Reviewer 1 Report
The authors did not address my concerns.
Reviewer 2 Report
The authors have answered my questions and made proper modifications to the manuscript. I don't have any concerns in publishing the paper given the manuscript is an introductory paper on FRA1 overexpression.
Reviewer 3 Report
The author's justification for not conducting functional experiments is not sufficient. The authors must provide some functional assays to support the conclusion of this article.